

# A new approach for increasing graduate students' science communication capacity and confidence

Erin R. Harrington[1], Scott R. McWilliams[1], Nancy E. Karraker[1], Caroline Gottschalk Druschke[2], Jenna Morton-Aiken[3], Elaine Finan[4] and Ingrid E. Lofgren[5]

[1] Department of Natural Resources Science, University of Rhode Island, Kingston, RI, USA
[2] Department of English, University of Wisconsin-Madison, Madison, WI, USA
[3] Department of English, Brown University, Providence, RI, USA
[4] Office for the Advancement of Teaching and Learning, University of Rhode Island, Kingston, RI, USA
[5] Department of Nutrition and Food Sciences, University of Rhode Island, Kingston, RI, USA

## ABSTRACT

There is an increasing demand for emerging scientists to improve their ability to communicate with public audiences, yet little research investigates the effectiveness of science communication training for graduate students. We responded to this need by developing SciWrite@URI—an interdisciplinary model for science graduate students designed around three learning outcomes based on tenets from the field of writing and rhetoric—habitual writing, multiple genres, and frequent review. SciWrite students completed courses and a science communication internship, attended writing workshops, and became tutors at a newly established Graduate Writing Center. After 2 years of training, students more frequently wrote multiple drafts and engaged in peer review, increased their confidence as writers, and decreased their apprehension about writing. We conclude the tenets of the SciWrite program helped students improve as science communicators, and we make suggestions for effective ways graduate departments and training programs might implement and build on our model.

## INTRODUCTION

### Why SciWrite?

Helping science graduate students build more effective writing habits, techniques, and confidence supports their graduate work and prepares them to meet the demands of a career in the sciences. A recent report by the National Science Foundation (NSF) and Council of Graduate Schools recommended the future of graduate education should include training for diverse career opportunities, and this training should include development of broad science communication skills (*Linton, 2013*). Most of the current emphasis on development of science communication expertise has been directed toward established scientists, both academic and non-academic, and not enough attention has

Corresponding author
Erin R. Harrington,
e_harrington@uri.edu

been focused on best practices for training graduate students in broader approaches to effective science communication (*Kuehne et al., 2014*). The traditional model of graduate student work involves first writing a brief proposal, and then completing milestones of data collection and analysis for research, with the majority of writing relegated to the end of students' academic programs. This delayed approach has consequences for research practice, for enhancing science competency of the broader public, and for science graduate students' future endeavors (*Kuehne et al., 2014*). Early engagement in scientific writing and science communication can contribute to students' ability to participate in their scientific disciplines (*Chinn & Hilgers, 2000*) and can improve students' comprehension of scientific concepts (*Keys, 1999*; *Wallace, Hand & Prain, 2004*). There is a demonstrable need for writing programs that engage graduate students in myriad forms of science communication from the outset of their graduate program. Such an approach will help students learn early and often how to best communicate their science through a number of genres to multiple non-specialist and non-academic audiences, as well as to their scientific colleagues.

The SciWrite@URI program (hereafter SciWrite)—initially funded through a NSF Innovations in Graduate Education grant—grew from the idea that barriers to graduate student participation in science communication and science writing include time constraints, lack of knowledge about opportunities, and a perceived lack of support from advisors (*Kuehne et al., 2014*). In addition, there is often disagreement among science faculty about the most helpful strategies for providing students with writing feedback and support, and many faculty attribute this confusion to a lack of adequate resources and/or inadequate departmental support in developing their own writing and teaching (*Pololi, Knight & Dunn, 2004*; *Reynolds et al., 2009*; *Kuehne et al., 2014*). Graduate students are writing under duress to meet a variety of important deadlines, and yet at many institutions, writing centers and writing classes only exist for undergraduate students. Furthermore, current approaches to writing feedback for graduate students often rely on individual mentors who may or may not have the knowledge necessary to provide useful feedback (*Florence & Yore, 2004*; *Odena & Burgess, 2015*) and thus access to writing support becomes an inequitable and unsustainable resource for many graduate students. SciWrite was designed by a community of practice to address many of these constraints by offering students and faculty with opportunities and institutional support for writing and communication. SciWrite responded to the reality that despite the recognized need for improved communication training for scientists, little research has been conducted on the most effective way to implement and assess such programs for STEM graduate students (*Heath et al., 2014*; *Kuehne et al., 2014*; *Druschke et al., 2018*; *National Alliance for Broader Impacts, 2018*; *Harrington et al., 2021*). SciWrite faculty members, representing the departments of Writing and Rhetoric, Nutrition and Food Sciences, and Natural Resources Science at the University of Rhode Island (URI), sought to create an innovative, interdisciplinary training model for graduate student writers that was both testable and replicable.

## Program objective and foundation

The aim of SciWrite was to create a unique training model that equips science graduate students with the tools necessary to be effective writers for multiple and diverse audiences. To provide students with practical, helpful writing tools, we used a rhetoric-based approach. Though rhetoric is often misunderstood as a term meaning "political spin," rhetorical studies is in fact a field of research dedicated to better understanding the ways in which we write and communicate (*Bitzer, 1968*; *Selzer, 2004*; *Druschke & McGreavy, 2016*). For instance, rhetoric allows us to understand how writing tasks vary depending on the genre and purpose, as well as the audiences' needs and expectations—this knowledge is crucial for any developing scientist who wishes to improve their writing and expand the reach of their scientific research (*Bazerman, 1988*; *Druschke & McGreavy, 2016*; *Druschke et al., 2018*).

Rhetorically-based writing instruction provides a number of best practices known to help students improve their writing and ultimately develop autonomy over their own writing and learning processes (*North, 1984*; *Straub, 1996*; *Coe, 2011*; *Petersen et al., 2020*). Using this rhetorically-based writing instruction framework, SciWrite created tangible and practical learning outcomes that graduate students engaged in during the program and could apply to their own immediate writing practice. At the time the grant was awarded, SciWrite was the first science communication program for graduate students that emerged as a collaborative effort between experts in life sciences and experts in writing and rhetoric to develop specific rhetoric-based learning outcomes, and we currently know of only one such other program (*King-Kostelac et al., 2022*). Furthermore, we know of very few science communication training programs that incorporate systematic and formal evaluation of student achievement of learning outcomes (but see *Baram-Tsabari & Lewenstein, 2016*, *2017*; *Rakedzon & Baram-Tsabari, 2017*; *Lewenstein & Baram-Tsabari, 2022*). We expected that this training would help participants continue to improve their writing even after completion of the program and would serve as a model for other institutions that wanted to do the same.

## METHODS

Three rhetorical tenets grounded the student learning outcomes of the SciWrite program: habitual writing (*Bruffee, 1981*), multiple genres for multiple audiences (*Porter, 1986*; *Gillen, 2006*; *Crowley & Hawhee, 2012*), and frequent review (*DiPardo & Freedman, 1988*; *Lunsford, 1991*; *Lundstrom & Baker, 2009*; *Druschke et al., 2018*; *Petersen et al., 2020*; *Harrington et al., 2021*, Fig. 1). From those tenets, we developed three primary learning outcomes for graduate students engaged in the SciWrite program (Box 1):

1) Habitual writing—students will produce high quality writing earlier and more frequently in their graduate school tenure
2) Multiple genres for multiple audiences—students will demonstrate effective command of writing in multiple genres for multiple audiences

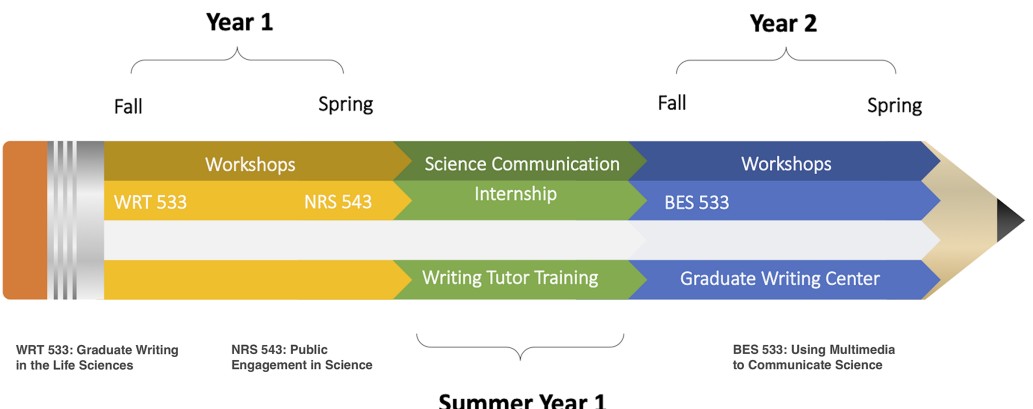

**Figure 1** Timeline of the 2-year SciWrite Program used to train each of the two cohorts of graduate students from 2016–2019: Progression through three courses (WRT 533 Graduate Writing in the Life Sciences, NRS 543 Public Engagement in Science, and BES 533 Using Multimedia to Communicate Science), simultaneous workshops, a summer science communication internship and writing tutor training and working as writing tutors at the URI Graduate Writing Center. Reprinted from *Harrington et al. (2021)* with permission.

3) Frequent review—students will evaluate peer drafts in order to provide helpful writing feedback and to improve their own writing skills

We also focused on emergent goals related to students' confidence as writers and scientists and their apprehension about writing (Box 1). We hypothesized that the key to increasing writing ability and confidence in SciWrite Fellows was this strategic development of learning outcomes based on common tenets from rhetorically-based writing instruction. We anticipated that although implemented for the program's local context, these rhetorically-informed writing practices are applicable to all disciplines and genres and should be able to transfer widely, to the variety of writing and communication situations that STEM graduate students face now and in the future (*Druschke et al., 2018*).

We designed program activities, workshops, and tutor trainings around our learning outcomes (Box 1). Assessment of learning outcomes incorporated a combination of self-reported surveys (Supplemental Appendix) and scoring of students' written products by writing assessment specialists. For the assessment of the learning outcomes related to habitual writing (LO1) and frequent review (LO3), we analyzed responses to a pre/post survey. The learning outcome related to multiple genres (LO2) was evaluated using a rubric that is used to assess quality and improvement in thesis/dissertation proposals and writing products from two graduate-level courses. These results were the focus of a separate article that demonstrated SciWrite fellows' writing scored higher on average than writing from students not formally engaged in the SciWrite program (*Harrington et al., 2021*). Results from LO2 are presented elsewhere (*Harrington et al., 2021*) and are discussed here only as context for the SciWrite program implementation.

## Program recruitment and timeline

In 2016, and again in 2017, science graduate students were recruited to be SciWrite Fellows *via* departmental, college, and university communications (email listservs, newsletters,

**Box 1 Program learning outcomes and emergent goals.** Learning outcome themes are: habitual writing, multiple genres for multiple audiences, and frequent review. Goal themes are: increased confidence as writers and scientists, and decreased apprehension about writing.

**Program Learning Outcomes**

**Learning Outcome 1. Habitual writing**

Students will produce high quality writing earlier and more frequently in their graduate school tenure

**Learning Outcome 2. Multiple genres for multiple audiences**

Students will demonstrate effective command of writing in multiple genres for multiple audiences

**Learning Outcome 3. Frequent review**

Students will be able to evaluate peer drafts in order to provide helpful writing feedback and improve their own writing skills

**Emergent Goals**

Goal 1. Students' confidence as writers will increase

Goal 2. Students' confidence as scientists will increase

Goal 3. Students' apprehension about writing will decrease

---

announcements, *etc*.), and faculty and staff recommendations. Candidacy required that students were URI graduate students with at least 2 years remaining in their program (M.S. or Ph.D.) and were a science major. We defined "science" using the NSF's broad definition for STEM disciplines which includes science, technology, engineering, mathematics, and social and economic sciences (*O'Meara et al., 2018*). We recruited a total of 12 Fellows, 75% of which were female ($n = 9$) and 25% of which were male ($n = 3$). 8.3% of Fellows identified as Asian ($n = 1$), 8.3% as Black or African American ($n = 1$), 66.7% as White ($n = 8$), and 25% identified as having an ethnicity of Hispanic or Latino origin ($n = 3$). Nine Fellows were native English speakers, while three fellows spoke English as a second language. Research protocols were approved by the University of Rhode Island Institutional Review Board (IRB # HU1516-009) and all participants completed the informed consent process and signed informed consent forms.

The SciWrite program timeline consisted of two writing-intensive science communication courses and a summer science outreach internship in the first year, followed by a paid writing tutor training course, a multimedia journalism class, and writing tutor employment in the second year (Fig. 1). URI graduate students who were not SciWrite Fellows also enrolled in the courses and attended some of the workshops, but only SciWrite Fellows completed the full suite of courses and workshops and were trained as writing tutors.

## Designing courses using best practices for writing instruction

To our knowledge, SciWrite is the first science communication program to systematically and formally assess student achievement of specific learning outcomes based on rhetorical tenets, and best practices were incorporated into the program to help students meet each of the three learning outcomes (*Petersen et al., 2020*; Table 1). For example, to support students' ability to meet the learning outcome related to habitual writing (LO1), SciWrite instructors used an approach called scaffolding, in which longer assignments were broken

up into simpler, shorter assignments, rather than entire drafts. This helped students slowly get comfortable with smaller chunks of writing and incrementally build up to a full, final draft of each assignment. Such an approach allowed students to take on writing projects that seemed less daunting and worth fewer points than a full writing project, thereby aiding students in developing the habit of writing early and often for their assignments (*Coe, 2011*; *Petersen et al., 2020*).

To meet the learning outcome related to multiple genres (LO2), we incorporated the important concept of the "rhetorical situation" into our training (*Bitzer, 1968*; *Druschke & McGreavy, 2016*). The rhetorical situation refers to the context within which scientists communicate their research to others. Important parts of this rhetorical situation are the audience they are communicating with, the expectations and needs of that audience, and the purpose for communicating their research. SciWrite Fellows practiced analyzing and engaging in different rhetorical situations by first creating a public engagement project within a SciWrite course, followed by an experiential learning science communication internship. Over the course of 2 years, Fellows practiced writing in multiple genres and engaged with a variety of audiences.

To help students meet the learning outcome related to frequent review (LO3), we incorporated two best practices from writing and rhetoric into our training: a focus on higher-order rather than lower-order concerns, and facilitative feedback rather than directive feedback (*Elbow, 1981*; *North, 1984*; *Neman, 1995*; *Straub, 1996*). Higher-order concerns deal with matters such as thinking about audience needs and expectations, developing clear arguments, and adhering to genre conventions, rather than sentence-level editing. Facilitative feedback, in contrast to directive feedback and sentence-level editing, is a best practice used by writing centers that encourages writers to consider that there are a variety of choices to make in their revision process (*Neman, 1995*; *Straub, 1996*; *Reynolds et al., 2009*). A more open-ended approach to feedback allows students to have autonomy over their own writing and revision process (*Neman, 1995*; *Straub, 1996*). Such writing practices are critical to exercise when science students must determine how best to convey their results in writing (*Groffman et al., 2010*; *Druschke et al., 2018*). As peer reviewers, SciWrite Fellows focused on higher-order concerns and providing facilitative feedback in their courses, during their training, and eventually in their work as writing tutors at our newly established Graduate Writing Center. This allowed students to practice giving and receiving peer feedback in a structured, holistic way.

The three required courses (Fig. 1) in the SciWrite program—'Graduate Writing in the Life Sciences', 'Public Engagement with Science', and 'Using Multimedia to Communicate Science' were all carefully designed with the three rhetorically informed program learning outcomes in mind (Box 1; Table 1).

The first course of the SciWrite program, 'Graduate Writing in the Life Sciences', was a graduate-level science writing course in the Department of Writing and Rhetoric that served all URI graduate students interested in learning more about scientific writing (*Druschke et al., 2018*). There were four scaffolded writing projects, each in a unique genre with varied audiences (Table 1). The second SciWrite course (Fig. 1), 'Public Engagement with Science', was offered through the Department of Natural Resources Science

**Table 1 Program activities and corresponding learning outcomes, best practices, and assignments.**

| Program activity[a] | Learning outcome focus[b] | Best practices | Assignments |
|---|---|---|---|
| Graduate writing in the life sciences course | 1, 2, 3 | Scaffolding, audience awareness, facilitative feedback focused on higher-order concerns | Journal analysis, public science writing critique, public science writing piece, and fellowship application statement |
| Public engagement with science course | 1, 2, 3 | Scaffolding, audience awareness | Public engagement project, public engagement project report |
| Using multimedia to communicate science course | 2 | Audience awareness | Multimodal storytelling, interviews, op-ed, TED talk |
| Science Communication Internship | 2 | Audience awareness | Varied by organization |
| Writing tutor training | 1, 2, 3 | Scaffolding, facilitative feedback focused on higher-order concerns | Weekly responses to group discussions, mock writing center sessions, personal tutoring philosophy |

**Note:**
[a] Timeline for these program activities is outlined in Fig. 1.
[b] Descriptions of these three learning outcomes are in Box 1.

(*Druschke et al., 2018*). The course centered around theoretical and practical aspects of public engagement with scientific research, policy, and management, with an emphasis on science communication (Table 1). Students' final public engagement projects were experiential learning projects based on the individual student's own research. These projects were created for a variety of different types of public audiences, including K-12 students, professional adults, and members of groups typically underrepresented in the sciences (*Druschke et al., 2018*). The third SciWrite course, 'Using Multimedia to Communicate Science', was a course in the Department of Journalism for upper-level journalism undergraduates and science graduate students (Fig. 1). Students learned about and developed capacity to communicate using a variety of science journalism genres, including news reports, coverage of STEM poster sessions, and public science lectures, interviews, and radio news stories (*Druschke et al., 2018*). Both SciWrite Fellows and non-SciWrite students were enrolled in all three courses, which were framed around the SciWrite learning outcomes.

## Experiential learning: science communication internship

In Year 2, Fellows applied communication skills gained in Year 1 to a science communication internship. With the support of a SciWrite Team Faculty mentor, SciWrite Fellows participated in a writing-intensive summer internship for an external organization related to their own field of research (Fig. 1). The general goal was that SciWrite Fellows would help an organization convey its science to public and specialized audiences. This internship helped students learn how to identify the needs and expectations of different audiences and adjust their approaches accordingly. This internship prompted Fellows to apply SciWrite course and workshop concepts in a practical, hands-on way (*Crone et al., 2011*; *Heath et al., 2014*) and also provided Fellows the opportunity to gain professional experience writing in multiple genres for diverse audiences. Fellows identified suitable

topics and media outlets with their internship hosts and engaged with their peers on developing and revising each piece (Table 2). Internships, funded by NSF, generally lasted 6 to 8 weeks but the duration was flexible depending on the science communication pieces being developed (all internships were approximately 150 h).

## Writing center tutor experience

SciWrite Fellows were trained as writing center tutors for approximately 3 months prior to serving as independent tutors in the Graduate Writing Center. SciWrite Fellows participated in rhetorically-focused writing center tutor training developed around the three program learning outcomes (Box 1; Table 1). The training incorporated a genre theory approach (*Troyan, 2014*) and writing assignments that required SciWrite Fellows to write multiple, scaffolded drafts as well as provide feedback to other SciWrite tutor trainees on these same writing projects (Table 1). The training incorporated many of the best practices previously mentioned: audience awareness, scaffolding, providing facilitative rather than directive feedback, and focusing on higher-order rather than lower-order concerns (*North, 1984*; *Neman, 1995*; *Straub, 1996*). Fellows were trained using a peer-to-peer framework common to most writing centers (*North, 1984*; *Straub, 1996*). SciWrite Fellows then joined other writing center tutors for five hour-long team tutoring sessions at the URI Undergraduate and Graduate Writing Centers. Lastly, SciWrite Fellows were observed by the Graduate Writing Center Trainer and Graduate Writing Center Coordinator and were given advice on how to improve their feedback strategies and how to develop their own personal tutoring philosophy to guide their tutoring approach.

The following year, URI's Graduate Writing Center was successfully created with tremendous support from the URI Graduate School, Office of the Provost, and Writing and Rhetoric faculty and graduate students, as well as faculty and staff from a variety of other departments and universities. Fellows from our second cohort, as well as one fellow from our first cohort, served as writing tutors alongside our Graduate Writing Center Tutor Trainer and Graduate Writing Center Coordinator.

## Program assessment

SciWrite Fellows completed a pre/post survey about achievement of learning outcomes and emergent goals just before the beginning of their program, and immediately after the end of their program (see Supplemental Appendix for surveys). Surveys were developed in collaboration with URI's Student Learning Outcomes & Assessment team and adopted survey methodologies traditionally used by URI's Office for the Advancement of Teaching and Learning. These surveys contained identical questions to assess self-perceived achievement of learning outcomes, including changes in writing confidence and writing apprehension. SciWrite Fellows answered most of the questions using Likert scales. For the writing apprehension portion of the survey, students were asked a total of 26 questions about their apprehension with writing, and all Likert scale scores (1, *strongly disagree*; 5, *strongly agree*) were added together for a total writing apprehension score (*Daly & Miller, 1975*; *Daly, 1978*; *Fischer & Meyers, 2017*; *Güler et al., 2017*; for the complete Daly and
**Table 2 Examples of host organizations and science communication pieces produced by SciWrite Fellows and distributed to the public and non-specialists through a variety of media.**

| Host organization | General theme of science communication pieces | Media used |
|---|---|---|
| American Parkinson Disease Association, Rhode Island Chapter | Healthy diet, weight loss | Newsletters, brochures, handouts |
| National Park Service, Padre Island National Seashore | Sea turtle research and conservation | Magazine article, brochure, website article |
| Metcalf Institute | Marine Disease and Climate Change | Website articles for public audiences |
| International Reptile Conservation Foundation Journal | Conservation and natural history of reptiles and amphibians | Website articles for academic and public audiences |
| Narrow River Preservation Association | Climate change and conservation | Brochure, handouts |

Miller Writing Apprehension questionnaire, see Supplemental Appendix). All 12 SciWrite Fellows completed the pre-survey; seven SciWrite Fellows completed the post-survey.

We compared changes in learning outcomes, writing confidence, and writing apprehension over time for all Fellows in the SciWrite program. Because all 12 participants completed the pre-survey, but only seven completed the post-survey, we compared population means (rather than individual scores) for each metric. We used t-tests to compare means between data that were normally distributed and non-parametric Kruskal-Wallis tests for data that were not normally distributed.

# RESULTS

Overall, despite small sample sizes, our results suggest that the majority of learning outcomes and emergent goals (Box 1) were met for all of our Fellows, and SciWrite Fellows' scores improved from the beginning of the program to the end (Fig. 2). For the learning outcome related to habitual writing (LO1), our results indicate students were more likely to adopt the practice of creating multiple drafts of their writing by the end of the program (Fig. 2, $p = 0.023$). Scores for the learning outcome related to frequent review (LO3) also increased from the beginning of the program to the end, though this trend was not statistically significant ($p = 0.097$). Students' confidence in their writing ability increased over time, and students reported a decrease in writing apprehension over time although this latter trend was not statistically significant (Fig. 2, $p = 0.017$ and $0.079$, respectively). Interestingly, students did not report a change in their confidence as scientists over time while in our program (Fig. 2). In general, the standard deviations of the learning outcome scores consistently decreased over time, suggesting the SciWrite training produced more consistency among students in how frequently they created multiple drafts (LO1) and shared their work with others (LO3) (Fig. 2).

# DISCUSSION

Our assessment results suggest that the SciWrite Fellows gained knowledge and abilities on our learning outcomes and achieved the majority of our emergent goals. Over the course of the 2-year SciWrite program, SciWrite Fellows more frequently wrote multiple drafts (LO1: related to habitual writing), their confidence as writers increased significantly, they
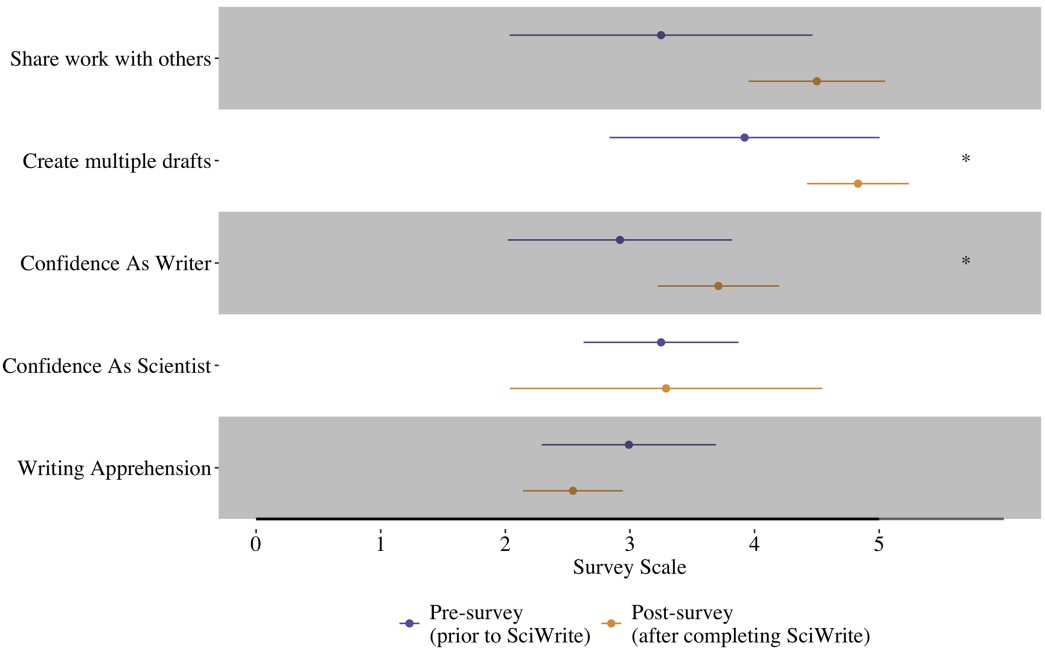

**Figure 2 Learning outcome and emergent goals survey scores for SciWrite Fellows comparing writing practices, confidence, and writing apprehension between the beginning and end of the 2-year program.** Points represent mean scores, Lines represent SD, and asterisks represent variables with significant *p*-values. Writing apprehension was rescaled to a 0-5 point scale.

tended to engage in peer review (LO3: related to frequent review), and their apprehension about writing tended to decrease. Though differences between pre- and post- survey scores were often modest, these modest changes could be partially due to the small sample size. Overall, these results indicate achievement of all learning outcomes and emergent goals, except for confidence as a scientist. In particular, the findings about writing multiple drafts, increased writing confidence, and decreased writing apprehension corroborate what other studies have found about writing apprehension and writing confidence: these two factors are often correlated, and they can negatively impact the student's motivation to write as well as the writing quality itself (*Daly & Miller, 1975*; *Daly, 1978*; *Pajares & Johnson, 1993*; *Fischer & Meyers, 2017*).

Students that participated in the SciWrite program gained additional confidence in their writing practice, likely due to the best practices from writing and rhetoric that we incorporated into our program. According to findings from other studies, using scaffolded writing assignments to encourage habitual writing, and engaging students in frequent review of others' work can help students improve not only in self-confidence (*Miller et al., 2015*) but also in critical thinking (*Coe, 2011*) and science communication (*Hryciw & Dantas, 2016*; *Petersen et al., 2020*). In addition, we suspect that because SciWrite Fellows became part of a community of practice (*Wenger, 1998*) when working at the Graduate Writing Center, this influenced our students' self-confidence as writers (*Kriner et al., 2015*; *Coffman et al., 2016*). This hands-on, community experience with receiving writing feedback from others and ultimately providing other science students with writing advice

allowed students to put what they learned into practice, and likely made SciWrite Fellows more comfortable with the writing process and thus more confident in their own abilities (*Cho & Cho, 2011*; *Aslan & Reigeluth, 2015*; *Cohen, Kulik & Kulik, 2016*; *Koh, Lee & Lim, 2018*). Anecdotally, many SciWrite Fellows mentioned that being a part of the Graduate Writing Center made them feel like they belonged to a writing community where they could support one another and give and receive writing advice, and this ultimately made them feel more confident in their own writing abilities and decreased their writing apprehension. A number of fellows also anecdotally reported giving and receiving writing feedback with each other beyond what was required for the SciWrite program. These reports align with related research which has shown that communities of practice can improve self-efficacy for graduate students (*Kriner et al., 2015*; *Coffman et al., 2016*; *O'Meara et al., 2018*).

There has been a growing interest in science communication training, with programs and courses focusing on communication training topics such as addressing the 'so what?', storytelling, and audience awareness (*McBride et al., 2011*; *Smith et al., 2013*; *Heath et al., 2014*; *Kuehne et al., 2014*; *Kloepper, 2017*; *Clarkson et al., 2018*). However, the URI SciWrite program is unique because of the emphasis placed on defined learning outcomes, the implementation of best practices using a rhetorical foundation, and the experiential learning that Fellows engaged in for their science communication internships and work in the Graduate Writing Center. To our knowledge, our science communication training program is the first program to systematically and formally assess student achievement of specific learning outcomes based on rhetorical tenets. We maintain that the commitment to strategically developing and measuring learning outcomes and implementing best practices common in writing and rhetoric courses to support student learning was key to SciWrite Fellows' learning and improvement. We suggest future programs build upon the approaches of SciWrite and further investigate potential best practices to support learning outcome achievement in science communication programs.

## Study limitations

Two limitations to our study were small sample size and lack of a "control" group of non-SciWrite Fellows. However, in a separately published manuscript, we report on an additional portion of our study in which we also evaluated non-SciWrite Fellows. *Harrington et al. (2021)*. In that study, we found that written work from SciWrite Fellows scored higher than those from non-SciWrite Fellows in all three genres, and most notably thesis/dissertation proposals were higher quality. Unfortunately, due to lack of funding and resources, our sample size was limited to 12 students. Despite these limitations, our major goal (program development), was performed robustly and our evaluation shows that students responded positively to the program that was designed to produce these outcomes. We believe this study provides the groundwork for future science communication programs to develop larger research projects that address additional research questions.

## Lessons learned and recommendations for future programs

SciWrite was a highly collaborative, interdisciplinary endeavor, and we believe that it is because of this collaboration that the legacy of the SciWrite program has continued for several years now past the period of the NSF-IGE grant. The research team included staff and faculty collaborations such as the URI Writing Center staff, faculty from the Department of Writing and Rhetoric, the College of the Environment and Life Sciences, and the College of Health Sciences, and the Director of Writing Across URI, a writing across the curriculum program. These interdisciplinary collaborations, coupled with the founding of the Graduate Writing Center, the development and launch of a Graduate Certificate in Science Writing and Rhetoric, and the creation of three new courses in science communication situated in several departments, resulted in SciWrite creating self-sustaining communities of writing practice for our graduate students and an infrastructure that remains and represents the legacy of SciWrite. Importantly, a final lasting effect of SciWrite is that science graduate students can now receive training very similar to that provided through the SciWrite program through enrollment in the Graduate Certificate in Science Writing and Rhetoric at URI, which includes completion of a set of science writing and communication courses, a science communication internship off campus, and the potential of completing tutor training and working in the Graduate Writing Center. Previous research has found such communities of practice are crucial for graduate student development (*Kriner et al., 2015*; *Coffman et al., 2016*; *Grainger et al., 2017*; *O'Meara et al., 2018*). Communities of practice are also particularly important when considering issues associated with a lack of systematic instruction for graduate students, and thus disparity in individual support and success (*Florence & Yore, 2004*; *Odena & Burgess, 2015*). Creating equitable infrastructures of support are crucial when developing writing/communication programs for graduate students, because many graduate students do not have access to the writing help they need and are solely relying on support from mentors who may or may not have the skills necessary to provide useful feedback (*Florence & Yore, 2004*). This inequity is of particular significance when considering research has found science publication to be a male and privileged space (*Macaluso et al., 2016*). SciWrite—thanks to support from the National Science Foundation and the faculty and key administrative leaders of our university—fundamentally and visibly changed the landscape of writing at URI. We recommend that future science communication programs develop similar strategies to provide infrastructure that will maintain interdisciplinary communities of practice to ensure the program has lasting impacts for students.

We believe that the Graduate Certificate in Science Writing and Rhetoric program that URI developed in collaboration with SciWrite can be used as a model for other science graduate departments lacking the funding and resources to develop a Science Communication training program. The Graduate Certificate in Science Writing and Rhetoric program adopted already existing courses in our English Department as well as various science departments, and worked with instructors to slightly modify the curriculum to incorporate our learning outcomes and best practices. For example, some best practices that are relatively simple for instructors to incorporate into already existing

courses are scaffolding and peer review (Table 1). Courses in the Graduate Certificate in Science Writing and Rhetoric program adopted an approach called scaffolding, in which longer assignments were broken up into simpler, shorter assignments, rather than entire drafts. This helps students slowly get comfortable with smaller chunks of writing and incrementally build up to a full, final draft of each assignment. Such an approach allows students to take on writing projects that seem less daunting and are worth fewer points than a full writing project, thereby aiding students in developing the habit of writing early and often for their assignments (Coe, 2011; Petersen et al., 2020). Courses in the Graduate Certificate in Science Writing and Rhetoric program also adopted a teaching tool known as "peer review" in which students practice giving and receiving peer feedback in a structured, holistic way. Science graduate departments looking to provide students with additional science communication training opportunities that don't require additional funding and resources can consult our Supplemental Materials and the Graduate Certificate in Science Writing and Rhetoric program's website (https://web.uri.edu/cels/academics/graduate-programs/certificate-in-science-writing-and-rhetoric/) for additional guidance.

## CONCLUSION

The program framework developed for SciWrite was purposefully designed in such a way that it could be easily adjusted and adapted to each institution's individual needs and goals. For example, our program was able to re-design courses that already existed within our university and that were taught by instructors with writing and rhetoric training. We recommend institutions wishing to adopt our program identify similar science communication courses in their own departments. Courses could then be modified to incorporate the best practices of shorter, scaffolded writing assignments, peer review practices, and audience awareness. We would also recommend that future programs collaborate with a variety of departments to develop a realistic implementation and assessment plan. We believe that what made the SciWrite program so effective was that we developed program learning outcomes rooted in rhetorical tenets and then provided students with the tools necessary to understand and apply these tenets in practical ways that they can now use in their day-to-day lives as scientists and communicators.

## ACKNOWLEDGEMENTS

We gratefully acknowledge the help and support provided by Cara Mitnick, Dr. Alycia Mosley Austin, Dr. Andrea Rusnock. We are indebted to Dr. Ashton Foley-Schramm for her hard work building and coordinating the Graduate Writing Center (GWC), and to Cara Mitnick for serving as the Administrative Director of the GWC. We are also incredibly thankful to Director Dennis Bennett and Coordinator Chris Nelson of Oregon State University's Graduate Writing Center for providing us with materials and advice as we developed the training program for our writing tutors. We would also like to thank Director Heather Price of URI's undergraduate Writing Center and Dr. Elena Kallestinova of Yale's Graduate Writing Center for their helpful counsel. We thank Dr. Sunshine Menezes for assisting with creating and implementing the program and Dr. Ali Scott for assessing initial data from the project. We would also like to thank Dr. Kendall Moore for

developing and teaching the third course of the SciWrite program (BES 533). In addition, we would like to thank Dr. Melissa Meeks of Eli Review for providing us with helpful advice and collaborating with us for numerous SciWrite workshops and projects. We of course also thank the courageous graduate students who were in the inaugural cohorts of SciWrite Fellows—without them none of this would have been possible.

### Funding

This research was funded by a National Science Foundation NRT-IGE Grant (#1545275) to Scott R. McWilliams, Ingrid E. Lofgren, Caroline Gottschalk Druschke, Nancy E. Karraker, and Nedra Reynolds. Additional funding was also provided by the URI Graduate School and URI's College of the Environment and Life Sciences. The funders had no role in study design, data collection and analysis, decision to publish, or preparation of the manuscript.

### Grant Disclosures

The following grant information was disclosed by the authors:
National Science Foundation NRT-IGE: #1545275.
URI Graduate School and URI's College of the Environment and Life Sciences.

### Competing Interests

The authors declare that they have no competing interests.

### Author Contributions

- Erin R. Harrington performed the experiments, analyzed the data, prepared figures and/or tables, authored or reviewed drafts of the article, and approved the final draft.
- Scott R. McWilliams conceived and designed the experiments, analyzed the data, authored or reviewed drafts of the article, and approved the final draft.
- Nancy E. Karraker conceived and designed the experiments, authored or reviewed drafts of the article, and approved the final draft.
- Caroline Gottschalk Druschke conceived and designed the experiments, authored or reviewed drafts of the article, and approved the final draft.
- Jenna Morton-Aiken conceived and designed the experiments, authored or reviewed drafts of the article, and approved the final draft.
- Elaine Finan conceived and designed the experiments, authored or reviewed drafts of the article, and approved the final draft.
- Ingrid E. Lofgren conceived and designed the experiments, performed the experiments, analyzed the data, authored or reviewed drafts of the article, and approved the final draft.

### Human Ethics

The following information was supplied relating to ethical approvals (*i.e.*, approving body and any reference numbers):

University of Rhode Island Institutional Review Board.

## Data Availability

The raw measurements are available in the Supplementary Files.

## Supplemental Information

Supplemental information for this article can be found online at http://dx.doi.org/10.7717/peerj.18594#supplemental-information.

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
