# Peer review of "A new approach for increasing graduate students’ science communication capacity and confidence"

_PeerJ, doi:10.7717/peerj.18594_

## Round 0.1 · original submission · Major Revisions

Your manuscript has been reviewed and assessed by three reviewers, and all of them agree that important points need to be addressed. The comments of the reviewers are included at the bottom of this letter. Reviewers indicated that results and discussion sections should be improved. I also recommend you clarify why you use non-parametric Kruskal-Wallis tests rather than the paired nonparametric test option. Additionally, please provide exact p-values in the results section. We would be glad to consider a substantial revision of your work, where the reviewer’s comments will be carefully addressed individually.

Reviewer 1 ·

Basic reporting

1. The biggest issue I see in the literature and background section is the conflation of "science communication" and "academic science writing". While academic science writing is certainly a component of science communication, it is not the only component. You switch from discussing science communication in line 24 (which would include communication with diverse non-scientific audiences) to just science writing to academic peers in line 29-30 to science writing to non-scientist audience in lines 33-35. This is confusing and needs to be better explained in the setup of this paragraph. Table 2 seems to indicate there are activities beyond just writing and academic writing - perhaps the naming of programming of SciWrite vs science communication and how you report these overlapping but not identical activities would be helpful.

2. In line 82 you cite Baram-Tsabari and Lewenstein 2016, 2017 - you should also cite Lewenstein and Baram-Tsabari's 2022 update on learning objectives for science communication training.

3. In terms of organization of the paper, much of the details about the course/intervention were included in the introduction rather than the methods. This was confusing as a reader.

4. Finally, there was ambiguity in the abstract language "students improved as writers by more frequently writing multiple drafts and engaging in peer review, increased their confidence as writers, and decreased their apprehension about writing." I don't understand if you mean that "writing multiple drafts and engaging in peer review" equates to "improving as writers" (I disagree with this claim) or if you mean "improved as writers by.... writing drafts, engaging in peer review, increasing confidence, and decreasing apprehension".

Experimental design

no comment

Validity of the findings

1. The paragraph beginning line 329 has lots of discussion content/explanation of reasons for results, rather than pure results. These editorializations should be moved to the discussion, and the results section should report just results.
2. Again, the paragraph beginning line 352 is entirely discussion and has no results. This should be moved to the discussion.

Additional comments

no comment

·

Basic reporting

See "Additional Comments"

Experimental design

See "Additional Comments"

Validity of the findings

See "Additional Comments"

Additional comments

General comments:
This paper describes a science communication training program that was developed to bridge identified shortcomings in STEM graduate training. The paper describes the development and elements of the 2-year program, which was informed by interdisciplinary pedagogical expertise, and built on rhetorical tenets. The authors evaluated the program effectiveness by pre and post program surveys.

The paper has many strengths and represents a unique contribution to the science communication and graduate training literature. These strengths are program development based on strong pedagogical theory, well defined outcomes, a corresponding robust program development around those defined outcomes and theory, and reasonably good evaluation. Weaknesses are namely the small sample size and that there is no control group for evaluation. However, I do not think these take away from the paper, which is clearly articulated with the goal of providing the template and justification for other departments or training programs to implement elements of the program to obtain and improve these learning outcomes.

Overall, the article is well written and easy enough to follow, but there is need for reorganization and revisiting the flow, and some sections could be strengthened. The Results and Discussion currently overlap in a way that is confusing. My specific comments below are intended to support revisions to address these.

Specific comments:
Line 11. Specify the programs – “graduate departments and training programs”?
Line 17. More like “supports their graduate work and prepares them to meet the demands of a career in the sciences”?
Line 20. Strong or broad science communication? I associate broad science communication skills with supporting diverse career paths, although of course just generally having strong skills helps with that. Perhaps consider how you talk about these aspects of communication – strong general research-writing skills vs. communicating for broad or diverse audiences – throughout the manuscript. It may be worth a line in your introduction to distinguish these different aspects, both of which might be considered scientific writing.
Line 29. Related to above, “scientific writing” could mean publishing peer-reviewed papers or broader communication (blogs, social media). I think it’s important to try and distinguish these for the reader, and specify which ones you are referring to at that time. For example, here you mean scientific writing as learning to write technical and/or peer-reviewed articles? If that is what is meant, it’s not well explained how research-based writing would lead to skills that support communication through “genres to multiple non-specialist and non-academic audiences”. I think my suggestion to try and address this is to spend 1-2 sentences in your Introduction defining what type of scientific writing this article is focused on, and/or what is encompassed (i.e., what approaches, training, or skills) in the term “scientific writing”, for the purposes of this article.

Line 46-47. I think delete “despite the fact that many graduate students lack adequate writing support”. It’s a difficult statement to support, and the point that writing centers are mostly available for undergraduates is sufficient.

Line 47-49. Right now this is phrased as though the community of practice was designed to address. Can be phrased better as “SciWrite was designed by a community of practice to address these constraints…”

Line 50-57. This is really well stated.

Line 60. Remove “better”

Line 61. Ah, I see, the goals are provide “tools necessary to be effective writers for multiple and diverse audiences” – so this is what you mean by “scientific writing”. Maybe bring in this same definition earlier on, would address my earlier comments.

Line 69. Is there any evidence or linkages between rhetoric-based training and effectiveness of research productivity, not only communication for broad audiences? Personally, I’ve found that science communication training (rhetoric training) makes me much more effective at proposal writing and journal articles, not only broad communication. But this may be a difficult thing to support and/or capture; it’s likely there isn’t research that supports this connection. If there were, a statement could be inserted here along the lines of “This type of training has also been shown to benefit and support scientists in their research-based activities…”

Line 86-89. It’s a little confusing and redundant to have the rhetorical tenets and the outcomes numbered. I’d remove the numbers in Lines 86-89, and just have the learning outcomes numbered (lines 92-96)

Line 105. Instead of “SciWrite students”, maybe “STEM graduate students”
Line 109. Learning outcomes being met by participants is awkward – maybe “whether student participants obtained or achieved learning outcomes”, or just “learning outcomes were met using a combination…”
Line 111. Outcome needs an ‘s’
Line 112-114. Don’t need to explain pre- and post-surveys, could just say “pre/post program surveys”.
Line 115. Right now reads as though the learning outcome was the development of the rubric. Suggest something like “The learning outcome related to…was evaluated using a rubric that is used to assess quality and improvement in thesis/dissertation…”
Line 117-119. I think you need to make more explicit how these results are or are not included. Up to this point, I was expecting reporting on all the outcomes, but then get confused what to expect at this point. Maybe just say something explicit here, like “Results from LO2 are presented elsewhere (Harrington et al. 2021) and are discussed here only as context for the SciWrite program implementation.”
Line 132. Remove “specific”
Line 137. Remove “in year 2”.
Line 141-142. Will be clearer if you merge these two sentences/ideas, as “based on rhetorical tenets, and best practices were incorporated…”
Line 144-145 and below. I don’t think you need to redefine the learning outcomes here, the short version and the acronyms (LO1 etc) are sufficient.

Line 155. “to help students learn how to write for public audiences” is run-on and somewhat redundant. Remove, break up, put elsewhere.

Line 181-183. This is more like a result or discussion – recommend moving there (probably discussion).

Lines 188-195. This also belongs in a discussion section. I think this section would read better if it skips to the description of the course (Line 197).

Line 204-208. This is a very long sentence

Line 214. Recommend “in all three courses, which were framed around the SciWrite..”

Line 217-219. This sentence should be substantially simplified. Maybe “In Year 2, Fellows applied communication skills gained in Year 1 to…”

Lines 231-273. Needs a section header “Writing Center Tutor Experience” etc. There are some sentences written twice at the start of this section too. Overall, however, I found this section lengthy and meandering. Some of it includes material for the Discussion (lines 239-241). I recommend carefully auditing and streamlining this section based on your goals – is it to communicate the (detailed) structure of the tutor program for someone that would establish one, or to describe it “well-enough” for the evaluation? Right now it seems to be aimed at the former, and feels excessively detailed.

Line 277-278. Delete - don’t need to define pre-post surveys.

Line 280-286. Just mention somewhere that the survey questions were Likert-scale, doesn’t need a whole sentence. Also, I would not say it is typical to describe the scales for different question types. These are standard scales, and the actual scales used are available in your supplemental materials and/or shown in results.

Line 291-295. You can just say “using paired t-tests or (for data that were not normally distributed) non-parametric Kruskal-Wallis tests”.

Line 297-312 – isn’t this Results?

Line 301-302. Remove predictions from Results

Under Results you have a sub-header, but there is only one (usually sub-headers are used when there are multiple sub-sections). Also, the text under Results is far more along the lines of Discussion, as it is interpretation of your Results. You could retain some of it, and merge with the reporting of figures and statistical results, but I think a lot of it is outside of what would be typical for Results. For example, you could keep Lines 316-321, and just support them by citing the figures and test results within. But move your interpretation (Lines 321-327) to Discussion. Right now, it seems like Lines 297-312 are describing the figures, and then the results and implications of the figures are (re-described) in text under Results…

Lines 329-350. These are all Discussion. But, as written, they are also basically a rehashing of the project design and hypotheses. For example, it seems strange to say that you suspect the improvement in learning outcomes resulted from program design, when the program was designed to improve learning outcomes. I think the information in this section should be re-written more along the lines of, “evaluation showed that students that participated in the SciWrite program gained…”. You can certainly include the anecdotal reporting too, that’s appropriate and interesting in a Discussion.

Lines 352-362 are well stated (but belongs in Discussion).

Lines 388-390. You say that “science communication programs” should work to implement these practices – but does that include science graduate departments too, or communication-specific programs only? Overall, I’d love to see additional specificity in your recommendations on how these practices and outcomes might be (feasibly) integrated into graduate training programs, given that most won’t necessarily be able to obtain separate grants. Given (often) low buy-in and resources for this type of training, are there ways to feasibly incorporate some of these structures and/or outcomes into a “standard” graduate training program? Perhaps by considering these learning outcomes being added to curriculum, adopting elements of SciWrite, creating and offering similar courses? Table 1 lays these options out nicely and your Conclusion heads in this direction, but stops short of where I think a strength of this paper lies, which does include the evaluation (albeit with small sample sizes), but I think more importantly, robustly defines rhetorical tenets and learning outcomes, and then designing training elements that support those. I recommend revising your Discussion to tie together specific elements in Table 1, the content in Lines 329-350 that I recommended move to Discussion, and your Conclusions more systematically. For example, you could reference how a department might adopt some of the specific activities from Table 1, piecemeal if they can’t do the whole thing, and give some examples.

Discussion overall. There is currently no inclusion of weaknesses or caveats in the evaluation, which is (namely) that that there was no evaluation of non-participants. So, a critic could easily say that these are skills that all students will tend to gain in early grad school years. It’s also a small sample size. I think it’s fine to simply acknowledge these issues and/or recommend additional evaluation, and emphasize that a major goal was program development, which was done robustly etc. And you can of course say that your evaluation shows that students responded positively to the program that was designed to produce these outcomes.

Figures 2-3. I'm confused about whether this is 2a and 2b or 2 and 3. It’s also very unclear which are learning outcomes and emergent goals – is 2b) only one emergent goal? I see, a) and b) are for the different survey scaled questions. Since you have questions on different scales, I suggest that you rescale them all to a 0-1 scale, so that you can show them on the same graph. And then you would just note in your figure caption that your responses were rescaled 0-1 for plotting. You could explain the min-max “values” for your different responses in the caption too (e.g., not at all confident (min) to quite confident (max).” I suggest this approach because I don’t think this graph needs to convey changes on the absolute scale(s) for these responses, but just show the overall directions and extent of change. What I mean is that you don’t need to exactly know that someone went from ‘somewhat confident’ to ‘quite confident’, rather that they improved in that dimension. And for Writing Apprehension, the scale isn’t meaningful anyway.

·

Basic reporting

Introduction: I greatly appreciated the framing of your article and your attention to graduate STEM science writing education.
- Important: While you mention the need for writing/communication programs, this claim would be better supported by mentioning the ways in which graduate students do typically learn to write and the flaws in the current/past approaches. For example, Odena and Burgess provide insight into thesis writing (https://www.tandfonline.com/doi/full/10.1080/03075079.2015.1063598); Florence and Yore, 2004 (https://onlinelibrary.wiley.com/doi/abs/10.1002/tea.20015) provide insight into science writing more specifically at different experience levels. Both of these groups illuminate the role of the mentor in supporting students, but also the lack of systematic approaches to support students. The lack of systematic instruction could easily be connected to equity in success, as current approaches rely on individual mentors who may or may not provide useful feedback. If graduate students have to learn by doing, or rely on mentors, then learning to write effectively is essentially based on access. Although not part of your study, I think this is something you could follow up on in your discussion as well. It is particularly important because science publication is a male and privileged space (Macaluso, Benoit, et al. 2016 https://pubmed.ncbi.nlm.nih.gov/27276004/ .

Beyond the comments above, the article is clearly written, referenced, and structured.

Experimental design

- Important: The authors do not provide the number or demographics of the participants, which I think is extremely important when determining the power of the programmatic intervention and general impact on various populations. Any information on participants should be included in the main manuscript.
- Important: The authors should include information on how the surveys were constructed and if they had any validation measures. For example, was there a focus group to examine how participants might interpret the questions?
- Minor- Figure 1: Please add the assessment points to this figure to help the reader better visualize when pre/post surveys were administered

Validity of the findings

- Important: I am confused at how Figure 2 relates to learning outcomes and emergent goals. For example, there are 3 emergent goals, but only 1 is shown in Figure 2. Therefore, how can the authors state that “results indicate overall achievement of all learning outcomes and emergent goals, except for confidence as a scientist” (lines 321-322). Please provide results for all learning outcomes and emergent goals.
- Please make sure all claims are supported by provided evidence. For example, lines 33-348 make claims of results that have no data associated with them.
Discussion:
- Important: the results are modest, but these modest changes could be partially due to the small sample size, please add this to the discussion or results.

---

## Round 0.2 · accepted · Accept

Thank you very much for submitting a revised version of your paper. I have gone through the revised, track-changes manuscript and rebuttal letter, and see that the authors addressed the reviewers' concerns and substantially improved the manuscript's content. So, based on my assessment as an academic editor, the manuscript may be now accepted for publication.

Reviewer 1 ·

Basic reporting

The clarity of organization was improved.

Experimental design

The edits on figures were useful.

Validity of the findings

No comment

Additional comments

No comment

·

Basic reporting

The revised manuscript has substantially improved the basic reporting, and I have only one comment, which is that Lines 420-427 contain nearly identical text/description of “scaffolding” that occurs in the Introduction. It would be better to reduce or remove these lines or replace them with something more informative.

Otherwise, I have no comments or further suggestion on the basic reporting.

Experimental design

The revised manuscript has addressed points that required clarification or streamlining , and I have no further comments or suggestions.

Validity of the findings

The revised manuscript has provided useful/missing points of context, including limitations of the study, and I have no further comments or suggestions.

Additional comments

The authors have done a comprehensive revision, and thoughtfully addressed reviewer comments; the resulting manuscript is clear, easy to read, and a useful contribution to the literature.

·

Basic reporting

The authors have adequately addressed all the previous reviewer comments on these points with one extremely minor exception.

Reviewer 1 had suggested a reference (Baram-Tsabari's 2022), which the authors now include as a citation in the introduction but did not include in the reference section.

Experimental design

The authors have adequately addressed all the previous reviewer comments on these points. The authors have substantially reworked the methods and results sections.

Validity of the findings

The authors have adequately addressed all the previous reviewer comments on these points. In particular, the authors now have a section on the limitations of the study which accurately contextualizes the findings.

Additional comments

Having addressed the original set of reviews thoroughly, I would recommend that this paper proceed to publication.